# Mutation of *Leaf Senescence 1* Encoding a C2H2 Zinc Finger Protein Induces ROS Accumulation and Accelerates Leaf Senescence in Rice

**DOI:** 10.3390/ijms232214464

**Published:** 2022-11-21

**Authors:** Chao Zhang, Ni Li, Zhongxiao Hu, Hai Liu, Yuanyi Hu, Yanning Tan, Qiannan Sun, Xiqin Liu, Langtao Xiao, Weiping Wang, Ruozhong Wang

**Affiliations:** 1Hunan Provincial Key Laboratory of Phytohormones and Growth Development, College of Bioscience and Biotechnology, Hunan Agricultural University, Changsha 410128, China; 2State Key Laboratory of Hybrid Rice, Hunan Hybrid Rice Research Center, Changsha 410125, China; 3National Center of Technology Innovation for Saline-Alkali Tolerant Rice in Sanya, Sanya 572000, China

**Keywords:** leaf senescence, ROS, cell death, LS1, rice

## Abstract

Premature senescence of leaves causes a reduced yield and quality of rice by affecting plant growth and development. The regulatory mechanisms underlying early leaf senescence are still unclear. The *Leaf senescence 1* (*LS1*) gene encodes a C2H2-type zinc finger protein that is localized to both the nucleus and cytoplasm. In this study, we constructed a rice mutant named *leaf senescence 1* (*ls1*) with a premature leaf senescence phenotype using CRISPR/Cas9-mediated editing of the *LS1* gene. The *ls1* mutants exhibited premature leaf senescence and reduced chlorophyll content. The expression levels of *LS1* were higher in mature or senescent leaves than that in young leaves. The contents of reactive oxygen species (ROS), malondialdehyde (MDA), and superoxide dismutase (SOD) were significantly increased and catalase (CAT) activity was remarkably reduced in the *ls1* plants. Furthermore, a faster decrease in pigment content was detected in mutants than that in WT upon induction of complete darkness. TUNEL and staining experiments indicated severe DNA degradation and programmed cell death in the *ls1* mutants, which suggested that excessive ROS may lead to leaf senescence and cell death in *ls1* plants. Additionally, an RT-qPCR analysis revealed that most senescence-associated and ROS-scavenging genes were upregulated in the *ls1* mutants compared with the WT. Collectively, our findings revealed that LS1 might regulate leaf development and function, and that disruption of LS1 function promotes ROS accumulation and accelerates leaf senescence and cell death in rice.

## 1. Introduction

Rice (*Oryza sativa* L.) is an important crop and one of the major food sources worldwide. It is the staple food for over half of the world’s population [1]. Leaves are the main organ of photosynthesis in plants. Early senescence of rice leaves affects photosynthesis and organic matter accumulation, ultimately leading to a significant decline in the yield and quality [2]. Leaf senescence is a highly complex and delicate process that is controlled by a precise molecular regulatory network [3,4]. Any disruption in the processes of this molecular regulatory network can lead to abnormal leaf senescence, which results in fluctuations in crop yields [4,5]. Leaf senescence is a normal physiological process in plants that occurs at a certain stage of their life cycle; it is accompanied by an orderly progression of some physiological and biochemical reactions and a redistribution of photosynthetic products [3,4,6]. However, the premature senescence of leaves is associated with the changes in intracellular physiological and biochemical features; this includes synthesis, degradation, and transportation of various proteins; degradation of macromolecules [4,7,8]; severe degradation of chloroplasts [8]; peroxidation of membrane lipids; and DNA damage [7,9]. Therefore, it is critical to understand the mechanism underlying premature leaf senescence for molecular breeding of crops.

Leaf senescence is a characteristic that is often induced by various internal factors (phytohormones, reproduction, reactive oxygen species (ROS), and some signal molecules), external or environmental factors (weak light or darkness, ozone, UV-B, and nutrient limitation), and biotic and abiotic stresses (mainly high or low temperatures, water, high salinity, oxidation, and pathogen attacks) [3,10,11,12,13]. Simultaneously, leaf senescence is accompanied by programmed cell death (PCD) caused by excessive ROS [14,15,16]. PCD is an essential process that determines plant growth and development and plays a critical role in the self-destruction of damaged cells [17,18]. In plants, organized destruction of cells is very important for the removal of infected and damaged cells and formation of specific organs [19,20]. Plants have evolved various protective mechanisms for disease resistance to prevent serious damage from pathogen attacks; the most common mechanism is the hypersensitivity response, which triggers rapid PCD to avoid further invasion in host plant tissues [10,21,22].

ROS are important signaling molecules that play an important role in cellular signaling pathways in response to diverse abiotic and biotic stresses [23]. In plants, ROS exist in different forms in many organelles such as chloroplasts, mitochondria, cytoplasm, and peroxisomes [24]. Many studies have reported that a high ROS content is a typical characteristic of senescent leaves; it is usually used as an index of successful leaf senescence, which was confirmed in *Arabidopsis* [25,26], wheat [27], and rice [16,28]. Excess accumulation of ROS such as superoxide anion (O^2−^) and hydrogen peroxide (H_2_O_2_) leads to an imbalanced redox state in cells and causes severe oxidative damage to intracellular substances [9,29,30]. To prevent excess ROS accumulation in cells, plants have developed an antioxidant defense system that includes ROS-scavenging enzymes: mainly ascorbate peroxidase (APX), superoxide dismutase (SOD), catalase (CAT), and peroxidase (POD) [30,31]. Moreover, various studies have reported that these ROS-scavenging enzymes are involved in the regulation of stress responses in plants [32,33,34].

Changes in leaf color or wilting are the most easily observed phenotypes in leaf senescence [35]. In recent decades, various types of mutants and several genes related to leaf senescence have been identified and characterized in plants [28,36,37,38,39]. In rice, cloned and functionally elucidated genes related to leaf senescence include transcription factors (TFs) such as the MYB/MYC and WRKY/NAC (NAM, ATAF1/2, and CUC2) TF families, zinc finger proteins (ZFPs), proteases, kinases, lipases, and ribonucleases [40,41,42,43,44]. Among these genes, *OsNAC2* (a rice NAC TF) participates in leaf senescence by activating the transcriptional levels of ABA biosynthetic genes (*OsZEP1* and *OsNCED3*) and inhibiting that of the ABA catabolic gene (OsABA8ox1) [40]. Moreover, OsNAC2 promotes the expression of chlorophyll-degrading genes (*OsSGR* and *OsNYC3*), which accelerates the degradation of chlorophyll [40]. OsWRKY42 (a WRKY TF) induces leaf senescence by repressing the mRNA levels of *OsMT1d*, a ROS-scavenging gene [42]. OsMYC2, a positive regulator of leaf senescence, activates the expression of some senescence-associated genes (SAGs) by selectively binding to their promoters [44]. Conversely, the MYB TF OsMYB102 represses the expression of SAGs and reduces ABA accumulation and ABA signaling responses, thereby delaying leaf senescence in rice [41]. OsTZF1, a CCCH-tandem type of ZFPs, delays leaf senescence by regulating stress-related genes such as *RD22*, *YSL6*, *akin-β*, and *AOS* [43]. In summary, various proteins, TFs, and other factors regulate leaf senescence via various pathways and constitute a highly complex molecular regulatory network.

Zinc finger proteins (ZFPs) constitute a large and diverse protein family in plants. Most of the ZFPs exhibit similar functions as TFs and are considered a special class of TFs [45]. They are involved in many processes of plant growth, development, and response to stresses. They play important roles in the functioning of cellular regulatory networks, including transcriptional regulation, DNA and RNA binding, and protein–protein interactions [46,47,48]. ZFPs contain one or more conserved zinc finger (ZnF) domains that are composed of relatively small protein motifs containing multiple finger-like protrusions [47,48]. Over the years, several ZFP genes related to leaf senescence have been cloned in rice, such as *OsDOS* [49], *OsTZF1* [43], *OsGATA12* [50], and *OsDOF24* [51]. In this study, we characterized rice leaf senescence 1 (ls1) mutants via a CRISPR/Cas9-mediated approach. *LS1* encodes a C2H2-type ZFP that was previously identified as stress response ZFP 1 (SRZ1) [52]. However, its function remains unclear. The *ls1* mutants exhibited leaf senescence, brown lesions on the upper and middle parts of flag leaves, a dwarf phenotype, and PCD. Further, we measured the ROS content and reported higher levels of H_2_O_2_ in *ls1* plants than in the wild type (WT) R893. Real-time quantitative PCR (RT-qPCR) analysis showed that several SAGs and ROS-associated genes were significantly upregulated in *ls1* plants. Furthermore, the induction of darkness accelerated the senescence of ls1 leaves compared with WT leaves. Our findings revealed that LS1 regulated leaf senescence and cell death through ROS metabolism in rice.

## 2. Results

### 2.1. Phylogenetic Tree and Expression Pattern Analysis

To investigate the subcellular localization of LS1 protein, we constructed a fusion expression plasmid of LS1-GFP and co-expressed it in rice protoplasts with the nuclear protein marker Ghd7-CFP. In protoplasts, the fusion protein LS1-GFP was predominantly observed in the nucleus and cytoplasm. The signals of LS1-GFP were co-localized with Ghd7-CFP, whereas GFP alone exhibited ubiquitous distribution throughout the cell (Figure 1A). These results revealed that LS1 is a nuclear- and cytoplasmic-localized protein.

According to the annotations of The Molecular Breeding Knowledgebase (http://mbkbase.org, accessed on 12 May 2020), the transcript of *LS1* was predicted to encode 166 amino acids. Moreover, to verify this prediction, we obtained the coding sequence of the *LS1* gene from the rice cultivar R893 (WT) using PCR and sequenced it. The sequencing data indicated that the length of the full-length CDS of the *LS1* gene was 501 bp and encoded a protein of 166 amino acids. Furthermore, using LS1 as the query in the NCBI database, 11 homologs were selected from various plant species (Figure 1B). The phylogenetic analysis indicated that LS1 and its homologs formed two distinct categories: one larger clade contained nine members, whereas the other clade had only three homologous proteins. Interestingly, the members of the larger branch were all from monocots, while the three homologs of other branch belonged to dicotyledonous plants (Figure 1B). These results suggested that LS1 had a closer relationship with other homologs from monocotyledonous plants such as *Zizania palustris*, *Sorghum bicolor*, or *Zea mays* than *Arabidopsis thaliana*, *Glycine max*, or *Gossypium*.

To characterize the sequence of all LS1 homologs, a sequence alignment and domain analysis were performed. The results revealed that LS1 and these homologs contained three highly conserved ZnF domains in tandem (I, II, and III; Figure 1C). In addition, LS1 shared higher amino acid sequence identity with proteins in monocot species such as *Panicum virgatum* (XP_039788691.1, 91.57%), *Setaria italica* (XP_004951522.1, 91.57%), *Dichanthelium oligosanthes* (OEL25621.1, 90.85%), *Sorghum bicolor* (KAG0532253.1, 89.76%), *Zizania palustris* (KAG8070361.1, 89.16%), *Miscanthus lutarioriparius* (CAD6251478.1, 89.76%), *Triticum dicoccoides* (XP_037453455.1, 78.21%), and *Zea mays* (NP_001132718.1, 79.52%) (Appendix A). LS1 displayed a high sequence similarity with its homologs in diverse species, which indicated that they may have similar functions. However, their functions are still unclear.

To investigate the expression patterns of *LS1* in different tissues and leaves at different stages, RT-qPCR assays were performed. The results showed higher expression levels of *LS1* in the panicle, stem, and leaf than that in the roots (Figure 1D). In addition, the results also showed that the expression level of *LS1* was lower in younger leaves and higher in mature or senescent leaves (Figure 1E).

### 2.2. Loss of LS1 Function Accelerates Leaf Senescence

To further explore the function of LS1 in rice, we conducted a targeted mutation of the *LS1* gene using CRISPR/Cas9-mediated technique. To efficiently and specifically mutate *LS1* in rice, a CRISPR/Cas9 vector containing two target sites (T1 and T2) was constructed (Appendix A). The binary vector pC-LS1-gRNA (Appendix A) was constructed as previously described by Ma et al. [53]. Further, the recombinant plasmid was introduced into the rice variety R893 via an Agrobacterium-mediated transformation. After the transformation, specific PCR and sequencing were performed to identify the positive transgenic plants. Further, two effective mutation types with different base insertions of 1 bp from T1 were selected and named as *ls1-1* and *ls1-2*, respectively (Figure 2A–C, Appendix A). The qPCR experiment showed that *LS1* was almost not expressed in *ls1* plants, which indicated that *LS1* gene was successfully knocked out in *ls1* mutants (Figure 2D).

The phenotypic characteristics of WT and *ls1* mutants that were grown in paddy fields in Changsha or Sanya with daily field management were investigated. The results revealed that the *ls1* plants exhibited premature leaf senescence, a reduced plant height, and slightly curled leaves at the heading stage (Figure 2A,B). At the seedling stage, the leaves of the *ls1* and WT plants exhibited no significant differences (Appendix A). Compared with WT, the leaf width and length were clearly lower in *ls1* mutants (Figure 2E,F). In addition, we detected the pigment contents in the leaves of WT and *ls1* plants. The results indicated that the carotenoid (Car), chlorophyll a (Chl a), and chlorophyll b (Chl b) contents in the senescent leaves of *ls1* plants were lower than those in the leaves of WT plants. Moreover, the total Chl content in the *ls1* mutants was significantly lower than that in the WT (Figure 2G). These results revealed that LS1 is involved in regulating leaf senescence during normal growth and development in rice.

### 2.3. Mutations in LS1 Lead to Increased Expression of Senescence-Related Genes

At the heading stage, the leaves of the *lsl-1* and *lsl-2* plants exhibited severe water loss, wilting, and brown lesions on the upper and middle parts of flag leaves compared with the WT (Figure 3A). To understand the mechanisms underlying the premature senescence of ls1 leaves, we examined the transcript levels of several SAGs [28], such as stay-green (*SGR*), *OsSAG12-2*, *Osl85*, *Os157*, *OsWRKY23*, and *OsNAP*, in the WT and *ls1* leaves using qRT-PCR assays. These data showed that the transcript levels of *SGR*, *OsSAG12-2*, *Osl85*, *Os157*, and *OsWRKY23* were significantly increased in the *ls1* plants compared with the WT. In contrast, there were no significant differences in the expression levels of *OsNAP* in the *ls1* and WT plants (Figure 3B). These results indicated that most SAGs were expressed at higher levels in the *ls1* plants than in the WT, suggesting that the process of leaf senescence was significantly accelerated in *ls1* leaves.

### 2.4. The LS1 Mutants Exhibited More ROS Accumulation and Cell Death

Many studies have reported that premature senescence in rice mutants was mainly induced by overaccumulation of ROS [54,55,56]. To investigate the ROS content in the *ls1* plants, histochemical staining was carried out to examine the levels of H_2_O_2_ and O^2−^ in the leaves (DAB, NBT, and TB staining for H_2_O_2_ accumulation, O^2−^ accumulation, and cell death, respectively). In the TB and NBT staining, more blue formazan precipitates were observed in the *ls1-1* and *ls1-2* leaves than in those of the WT (Figure 4A–C). In the DAB staining, the leaves were stained dark brown in the *ls1* plants; however, the leaves of the WT exhibited almost no staining (Figure 4D).

Intracellular accumulation of H_2_O_2_ accelerates leaf senescence or cell death. It was reported that the content of malondialdehyde (MDA) is a reliable indicator that reflects the damage to cells. Therefore, the contents of H_2_O_2_ and MDA were measured in the *ls1* and WT leaves. The results revealed that the H_2_O_2_ and MDA contents were significantly higher in the *ls1-1* and *ls1-2* leaves than in the WT leaves, which was consistent with the results of the DAB staining (Figure 4E,F). SOD catalyzes the conversion of O^2−^ into H_2_O_2_ and O_2_, and CAT accelerates the decomposition of H_2_O_2_ [57,58]. Thus, the SOD and CAT activities were measured in the *ls1-1*, *ls1-2*, and WT plants. The results indicated that the SOD activity was significantly increased and the CAT activity was clearly decreased in the *ls1* plants compared to the WT (Figure 4G,H). These results suggested that ROS overaccumulation and cell death occurred in the ls1 plants.

### 2.5. Loss of Function of LS1 Causes DNA Damage and Triggers PCD

Excessive accumulation of ROS such as H_2_O_2_ and O^2−^ may cause cellular damage or trigger PCD [59]. It is well known that PCD is a representative feature of rice leaf senescence [60]. To evaluate the PCD process in the *ls1* leaves, we performed a terminal deoxynucleotidyl transferase dUTP nick end labeling (TUNEL) assay to investigate the degree of DNA fragmentation. When the cells underwent apoptosis, the DNA broke and fluorescein could be added to the exposed 3′-OH, which could be easily detected using fluorescence microscopy. The results showed that TUNEL-positive signals were clearly detected in the leaves of the *ls1-1* and *ls1-2* plants, whereas almost no green fluorescein signals were detected in the leaves of the WT (Figure 5). These results suggested that severe DNA damage occurred in the *ls1* plants that further led to PCD.

### 2.6. Alteration in mRNA Levels of ROS-Related Genes in LS1 Plants

The balance of ROS in the plant body is closely related to the transcription of SOD, POD, and ascorbate oxidase (AO) genes. A growing stream of research reported that ROS-associated genes were induced by the excessive accumulation of ROS. For a further analysis of the molecular basis of the overaccumulation of ROS in *ls1* plants, we determined the mRNA levels of several ROS-related genes such as *AOX1a*, *AOX1b*, *APX1*, *APX2*, *APX8*, *SODB*, *SODA1*, *CatA*, and *CatB*, which strictly regulate the levels of ROS in plant cells [30,31]. The qRT-PCR results showed that the transcript levels of most genes including *AOX1a*, *AOX1b*, *APX2*, *SODB*, *SODA1*, and *CatA* were significantly upregulated in the ls1 plants, while no significant alterations of the transcript levels of APX1, APX8, or CatB were detected in the WT, *ls1-1*, and *ls1-2* plants (Figure 6). Therefore, most ROS-related genes exhibited higher expression levels in the ls1 plants, which were correlated with the overaccumulation of ROS.

### 2.7. Acceleration of Dark-Induced Leaf Senescence in LS1 Plants

A decrease in pigment content is one of the typical characteristics of leaf senescence. Dark-induced senescence, which is an effective method to simulate synchronous plant senescence, has been widely used in numerous studies on leaf senescence [61]. Accordingly, we evaluated the effects of dark-induced leaf senescence in the WT, *ls1-1*, and *ls1-2* plants. After 4 days of dark treatment, detached leaves from the *ls1-1* and *ls1-2* plants became more yellow than those from the WT plants and exhibited a higher rate of senescence in the mutants than that in the WT (Figure 7A). Moreover, we measured the pigment content in the WT and two *ls1* mutants after dark-induced senescence. Consistent with the yellow phenotype, the Chl a, Chl b, and Car contents were significantly reduced in the *ls1* leaves compared with the WT leaves (Figure 7B). These results suggested that disruption of the LS1 function accelerates leaf senescence upon the induction of darkness.

## 3. Discussion

### 3.1. Mutation of LS1 Leads to Leaf Senescence in Rice

Leaf senescence is an inevitable stage of annual crop plants. However, premature senescence causes a severe decrease in the yield and quality. Recently, some leaf-senescence-associated mutants of rice were isolated and characterized. The identification of these mutants contributed to the further exploration and understanding of the molecular mechanisms underlying leaf senescence. The dwarf and *early-senescence leaf1* mutant (*del1*) exhibited a decreased content of Chl and melatonin, tiller number, root length, plant height, and thousand grain weight [62]. The lower leaf tips and leaf margins of the progeria mutant *es4* turned yellow (mainly at the tillering stage), whereas the whole leaves of *es4* turned yellow and senescent at the grain-filling stage [63]. The *early senescence 2* (*es2*) mutant exhibited rapid leaf senescence; its leaves exhibited some yellow spots at the seedling stage and gradually withered at the tillering stage until maturity [61]. The mutant Nature Blight Leaf 1 (*nbl1*) exhibited leaf senescence and delayed growth compared with the WT at various developmental stages. At the seedling stage, the *nbl1* mutant exhibited delayed growth, and the lower leaves of the nbl1 mutant displayed distinct senescence. At the tillering stage, the *nbl1* mutant displayed yellowish and whitish leaf tips [28]. Although many leaf-senescence-associated genes have been cloned and identified to understand the process of leaf senescence, some gaps still exist in understanding the process in detail. In this study, *ls1* plants exhibited premature leaf senescence, decreased chlorophyll contents, a short plant height, small leaves, and cell death. Therefore, LS1 not only regulates leaf senescence but also participates in the regulation of other traits during growth and development. These results suggested that *ls1* is a mutant different from those reported above. First, the leaves of the *ls1* mutant displayed no clear differences compared with those of the WT at the seeding stage. Second, some brown spots appeared on the upper part of the *ls1* leaves from the tillering stage to the heading stage that became more severe at the heading stage. These results indicated that *ls1* is an ideal material to understand the mechanisms of leaf senescence.

In this study, we found that disruption of the LS1 function resulted in a phenotype of premature leaf senescence. *LS1* encoded a ZFP with an undefined function with three tandem-repeat ZnF domains. The phylogenetic tree revealed that LS1 was more closely related to monocot plants. Thus, we speculated that LS1 and its homologues may play a similar role. More than 10 years ago, a study reported that the expression of LS1 was regulated by various stresses; however, its function still had not been reported yet [53]. To our knowledge, this was the first study to report that LS1 plays an important regulatory role in plant growth and development, particularly in the regulation of leaf senescence. For further understanding of the mechanism of LS1 in regulating leaf senescence, the transcript levels of SAGs were investigated. Among them, stay-green (*SGR*), *OsSAG12-2*, *Osl85*, *Os157*, *OsWRKY23*, and *OsNAP* were the main marker genes of leaf senescence in rice [28,61]. The alterations in the expression levels of these genes reflected the senescence process of leaves to a certain extent. The qPCR results showed that the expression levels of most SAGs were significantly higher in the *ls1* plants than that in the WT. Senescent mutants typically displayed increased ROS accumulation and a higher expression of senescence-related marker genes [64]. Moreover, excessive ROS accumulation and severe cell death were detected in the mutants. Collectively, these results suggested that the *ls1* plants exhibited typical senescence features and that the process of leaf senescence was significantly accelerated in the *ls1* leaves. In addition, we found that leaf segments yellowed faster and the Chl content decreased more rapidly after incubation in the dark in the *ls1* plants than that in the WT. These results were consistent with studies on previously reported mutants such as *nbl1*, *osgst4*, and *oswrky5-D* [28,37,65]. Therefore, the disruption of LS1 function led to premature leaf senescence in rice under normal growth conditions and accelerated dark-induced leaf senescence.

### 3.2. Disruption of LS1 Function Leads to ROS Accumulation and Cell Death in Rice

Previous studies have reported that ROS are among the critical signaling molecules in cells; however, overaccumulation of ROS, particularly H_2_O_2_, can accelerate the leaf senescence process [66]. H_2_O_2_ is mainly produced in various organs via cellular metabolic pathways; its intracellular balance is mainly regulated by the antioxidant enzymes SOD and CAT [67,68]. Our data indicated that ROS content in the *ls1* leaves was remarkably increased compared with the WT as confirmed by several histochemical staining assays and H_2_O_2_ content measurement. The SOD activity increased and CAT activity decreased, although the qRT-PCR analysis revealed that the transcript levels of ROS-scavenging-related genes were higher in the *ls1* plants, which suggested that considerable levels of ROS accumulated in *ls1* leaves. There was lower activity of CAT in the *ls1* mutants, which suggested that ROS detoxification in these mutants was lower than in the WT plants. The increased ROS content induced the expression of CAT-related genes, but its enzyme activity was not enough to rapidly increase at a specific stage. Similar results were obtained by Zheng et al. in the leaf senescence mutants *msl-1* and *msl-2* [69]. Thus, we speculated that aberrant changes in the activity of two key enzymes that scavenged excess ROS resulted in an inability to timely remove ROS from cells, which led to an overaccumulation of excess ROS. Collectively, we speculated that *LS1* is involved in ROS metabolism and that disruption of *LS1* function leads to high levels of ROS in rice. However, excessive H_2_O_2_ production often induces PCD or cell death [14,15,16]. To investigate whether *LS1* was involved in ROS-mediated cell death in rice, histochemical strains and a TUNEL assay were used for further analysis. All of the staining results demonstrated that the mutation of *LS1* may trigger cell death in rice. The TUNEL assay indicated that the *ls1* leaves exhibited strong TUNEL-positive signals. Collectively, these results suggested that high cellular ROS levels were accumulated in the *ls1* leaves, which ultimately led to PCD or cell death. Currently, the mechanism underlying ROS-mediated leaf senescence and cell death is being elucidated. Cui et al. [9] revealed that mutations of *EARLY LESION LEAF 1* (*ELL1*) promoted ROS accumulation and cell death in rice. Yang et al. [28] reported that the disruption of *ES2* function accelerated ROS accumulation in *es2* leaves, which led to leaf senescence in rice. Zheng et al. [69] revealed that the mutation of *CYP71P1* promoted early senescence and cell death, mainly due to higher ROS levels in the mutant leaves. In this study, we demonstrated that a novel ZFP LS1 is involved in leaf senescence and cell death through ROS metabolism. Our study provided a basis for further understanding the molecular mechanism underlying ROS-mediated leaf senescence in rice.

## 4. Materials and Methods

### 4.1. Plant Material and Growth Conditions

Seeds of the R893 rice variety (*Oryza sativa* L. ssp. *indica*) were provided by the Hunan Hybrid Rice Research Center (HHRRC) as the WT seeds for the physiological experiments and genetic transformation. The R893 and *ls1* mutants were cultured at 28 °C in a greenhouse under a 14 h day/10 h night cycle or were grown in the field of the HHRRC in Changsha or Sanya with daily field management. Mature seeds of the R893 and *ls1* plants were harvested, dried, and stored in a refrigerator at 4 °C.

### 4.2. CRISPR/Cas9 Vector Construction, Rice Transformations, and Generation of Cloned Lines

The plant expression plasmids of pYLCRISPR/Cas9 and gRNA were provided by the team of Yaoguang Liu of South China Agricultural University. The sites containing (N)20GG or G(N)20 GG were confirmed as the target sequences in the coding or genome sequence regions of the *LS1* gene according to a previously described method [53]. Then, we designed sgRNA based on the two target sites according to the method described in [70]. Briefly, the specific primers containing the two (T1 and T2) target site sequences were ligated into the respective pYLsgRNA-U3 and pYLsgRNA-U6a cassettes. The overlapping PCR reaction was performed to construct the pYLCRISPR/Cas9 vector using the F-U/R-gRNA primers and site-specific F-B1/R-B2 and F-B2/R-BL primers. The pYLCRISPR/Cas9-LS1-T1/T2 recombinant plasmids were transformed into Agrobacterium tumefaciens EHA105. The rice transformation was performed as described previously [71]. After transformation, all of the transgenic seedlings were cultivated in the greenhouse or fields for further identification. To determine the mutation at the target sites, genomic DNA was extracted from approximately 30 mg of leaf tissue using a DNA Plant Kit (TsingKe Biotech, Beijing, China). The isolated and purified genomic DNA was used for PCR amplification using I5-TM (TsingKe Biotech, Beijing, China), and the fragments were amplified using specific primers (LS1-F/LS1-R) containing the two CRISPR/Cas9 target sites. Further, these PCR fragments were purified and sequenced to identify any *LS1* mutations. The sequence alignment analysis was performed using DNAMAN7.0 and MAGE6.0 software. The specific primers mentioned above are listed in Appendix A.

### 4.3. Histochemical Marker Staining Assay

The 3,3′-diaminobenzidine (DAB) and nitro blue tetrazolium (NBT) staining were performed to detect H_2_O_2_ and O^2−^ accumulation, respectively, as described previously with a small modification [72,73]. Briefly, the leaves of *ls1* mutants and WT were placed in 1% Triton X-100 solution for 15 s and transferred to a DAB (pH 3.8) or NBT (pH 7.8) solution followed by gentle shaking in the dark for 20 h at 28 °C. Further, the treated leaves were placed in 90% ethanol for 48 h until the chlorophyll was removed and then further placed in 75% glycerol for photographing. Trypan blue (TB) staining was performed as previously described for detecting dead cells [74]. Briefly, the leaves were dipped in boiling water for 2 min and then placed at room temperature for cooling. Further, they were soaked in TB staining solution with gentle shaking in the dark for 48 h and transferred to 75% glycerol for rinsing and photographing. Each treatment was repeated three times.

### 4.4. Dark Treatment

Dark-induced leaf senescence was conducted as described previously with some modifications [28,37]. In brief, the flag leaves of the *ls1-1*, *ls1-2*, and WT plants were cut into approximately 2 cm fragments at the tillering stage. Then, the detached fragments were placed in 15 mL ddH_2_O in petri dishes and incubated at 28 °C in complete darkness. After 4 days, the leaf fragments were used to determine the pigment content and were photographed. Each treatment was repeated three times.

### 4.5. Analysis of H_2_O_2_, MDA Content, and Enzyme Activity

At the heading stage, the leaves of the *ls1* mutants and WT grown in rice fields were used to prepare tissue homogenate for evaluating the CAT, H_2_O_2_, SOD, and MDA contents as described previously [75]. All procedures were performed according to the manuals for reagents of Nanjing Jiancheng Bioengineering Institute. Each treatment was repeated three times.

### 4.6. Determination of Photosynthetic Pigment Content

Leaves that were subjected to dark-induced senescence for 4 days were used to measure the chlorophyll content. Leaves without any treatment served as the control. The contents of Chl a, Chl b, and Car were measured using a method described previously [76]. Briefly, the leaves were cut into small fragments with a length of approximately 0.5 cm and a weight of 0.08 g, placed in 80% acetone, and soaked for more than 24 h in the dark with shaking every 5 h. Then, the absorbance of the supernatant of the tissue homogenate was measured at 645, 470, and 663 nm against 80% acetone as the blank using a microplate reader. The pigment content was calculated using the following formulae:Chl a = (12.7 × A663 − 2.69 × A645) × V/W;
Chl b = (22.9 × A645 − 4.68 × A663) × V/W;
Total Chl = Chla + Chlb;
Car = (1000 × A470 × V/W − 3.27 × Chla − 104 × Chlb)/198.
where V is the total volume of the chlorophyll extract (mL) and W is the fresh weight of the material (g). Each treatment was repeated three times, and a *t*-test was conducted in the statistical analysis.

### 4.7. TUNEL Assay

To examine whether PCD occurred in the *ls1* plants, we performed a TUNEL assay. Concisely, the WT and *ls1* leaves were cut into thin slices, fixed with 70% FAA fixative, and embedded in paraffin. The TUNEL staining was performed using a TUNEL assay kit as previously described [77]. Apoptosis was detected using a DeadEnd Fluorometric TUNEL system (Promega, #G3250, USA) according to the manufacturer’s instructions.

### 4.8. Subcellular Localization

To further determine the subcellular localization of LS1, we used the full-length *LS1* coding sequence to construct the pCA1301-LS1-GFP recombinant using the specific primers LS1-GFP-F/R (Appendix A). Briefly, the recombinant (35S: LS1-GFP) and nuclear marker (Ghd7-CFP) and similarly, the control vector (35S: GFP) and nuclear marker plasmid (Ghd7-CFP), were co-transformed into rice protoplasts for expression. For the transient expression, protoplasts were isolated and prepared from the leaf sheaths of the R893 rice cultivar at the seeding stage as described previously [78]. The transformed protoplasts were incubated at 28 °C for 20 h under weak light conditions. After transformation, the positively co-transformed protoplasts were selected, and GFP or CFP signals were detected using confocal laser scanning microscopy (Zeiss, #LSM880, Germany).

### 4.9. Phylogenetic Analysis and Sequence Alignment

The sequences used for the phylogenetic analysis were searched using the NCBI Blastp search program (http://www.ncbi.nlm.nih.gov, accessed on 18 June 2021) with the LS1 protein sequence as the query; the resulting protein sequences were then analyzed. The phylogenetic analysis of LS1 from various eukaryotes was performed using the maximum-likelihood method. A neighbor-joining tree was constructed using MEGA 6 software with the bootstrap method and 1000 replicates as previously described [79]. Multiple sequence alignments were performed using the software DNAMAN 7. The accession numbers of the proteins used in the phylogenetic tree construction are listed in Appendix A.

### 4.10. RNA Extraction and Quantitative Real-Time PCR

The total RNA was extracted from the leaves of the WT and *ls1* plants using a TRIzol reagent kit (Invitrogen, Carlsbad, CA, USA) according to the manufacturer’s instructions. To remove the genomic DNA, the RNA was treated with DNase I for 15 min. The cDNA was synthesized using TransScript^®^ First-Strand cDNA Synthesis SuperMix (Trans, #AT301-02, Bengjing, China). The specific primers used for the RT-qPCR were designed using Primer 5.0 software according to the transcript sequences of genes in Shuhui498. *OsActin* (*OsR498G0306938500*) was used as the reference gene. The RT-qPCR was performed using an ABI QuantStudio 3 Real-time PCR Instrument (ABI, USA) with 10 μL of PCR reaction mixture that included 0.5 μL of cDNA, 5 μL of 2 × SYBR Green Pro HS mix, 0.3 μL of forward primer, 0.3 μL of reverse primer, and 3.9 μL of nuclease-free water. The relative expression levels of genes were calculated using the 2^−ΔCT^ and 2^−ΔΔCT^ method. Three biological replicates were performed for each sample, and a *t*-test was used in the statistical analysis. The specific primers mentioned above are listed in Appendix A.

## Figures and Tables

**Figure 1 ijms-23-14464-f001:**
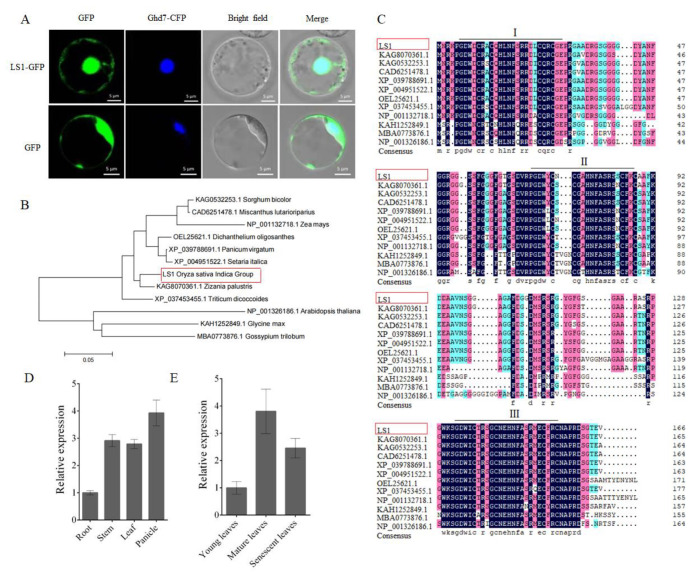
Expression assay and bioinformatic analysis of *leaf senescence 1* (*LS1*). (**A**) Subcellular localization of LS1. LS1 fused with green fluorescence protein (GFP) was transiently expressed in rice protoplasts. The bottom panels show the localization of GFP as the control. Scale bar = 5 μm. (**B**) Phylogenetic tree analysis of the LS1 and its homologies. Each accession number represents a gene; their information is available in NCBI. (**C**) Multiple sequence alignment analysis of LS1 and its homologies. (**D**) Expression pattern of *LS1* in various tissues. (**E**) Expression pattern of *LS1* in the leaves at various developmental stages. The error bars indicate SDs of three biological replicates (**D**,**E**).

**Figure 2 ijms-23-14464-f002:**
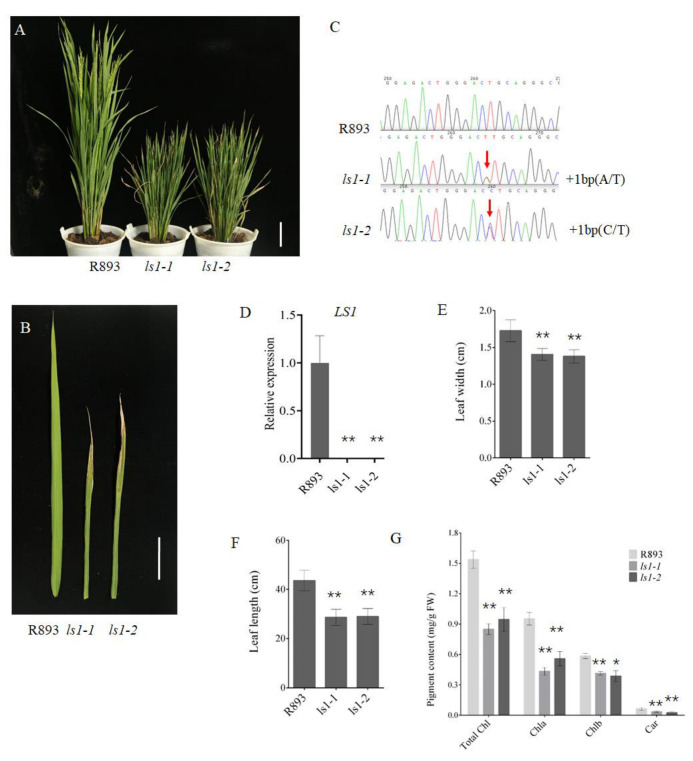
Phenotypic characterization of WT and *ls1* plants. (**A**) Phenotypes of WT and *ls1* mutants at the heading stage. (**B**) Leaf phenotype of WT and *ls1* plants at the heading stage. (**C**) Mutation sites of *LS1* gene in the *ls1-1* and *ls1-2* plants. The “+” indicates the insertion and the numbers indicate the number of bases. The red arrow indicates the insertion position of the coding region. (**D**) *LS1* expression level in WT and *ls1* plants. (**E**,**F**) Length and width of leaves from WT, *ls1-1*, and *ls1-2* plants. (**G**) Determination of photosynthetic pigment content. The error bars indicate SDs of three biological replicates. The *p*-value was calculated using a Student’s *t*-test. ** *p* < 0.01, * 0.01 < *p* < 0.05. Bar = 20 cm (**A**); bar = 10 cm (**B**).

**Figure 3 ijms-23-14464-f003:**
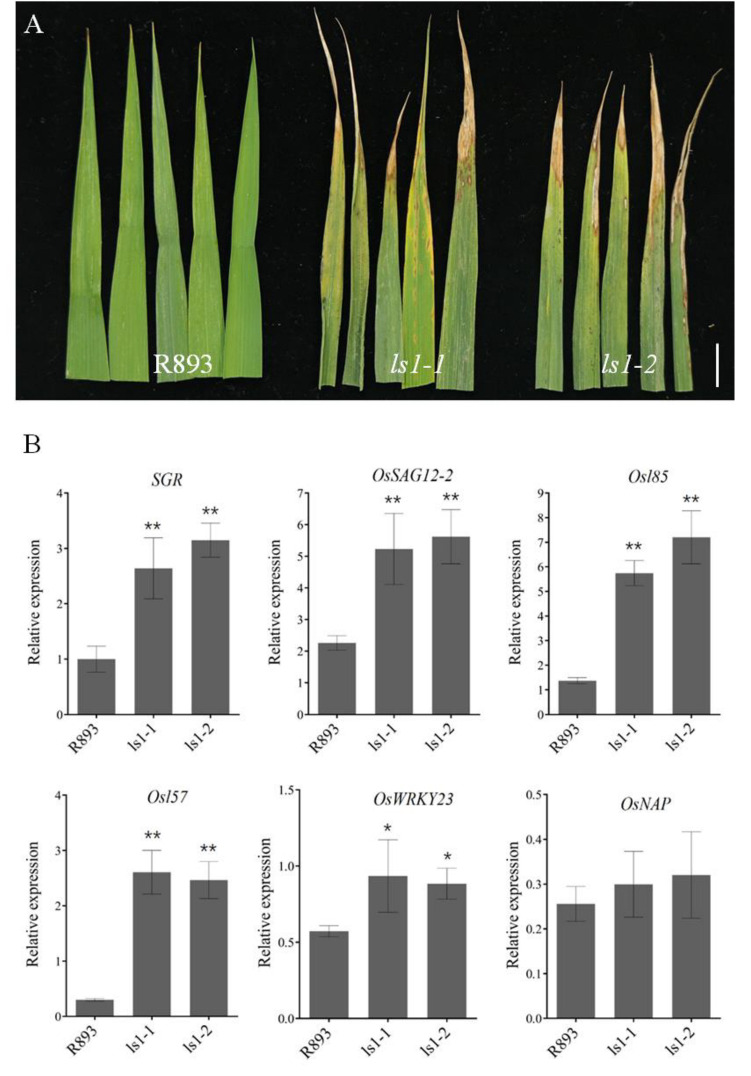
Leaf upper phenotype in *ls1* plants at the heading stage and expression levels of SAGs. (**A**) Leaf phenotype of WT and *ls1* mutants at the heading stage. (**B**) Transcript levels of six SAGs were analyzed using RT-qPCR. The error bars indicate SDs of three biological replicates. The *p*-value was calculated using a Student’s *t*-test. ** *p* < 0.01, * 0.01 < *p* < 0.05. Bar = 2 cm (**A**).

**Figure 4 ijms-23-14464-f004:**
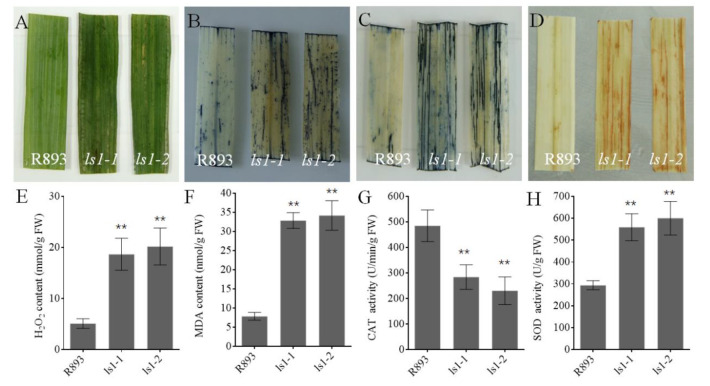
Analysis of ROS contents and measurement of physiological and biochemical parameters. (**A**) The leaves of WT, *ls1-1*, and *ls1-2* plants with no treatment at the heading stage. (**B**–**D**) NBT-, TB-, and DAB-stained leaves of WT and *ls1* plants. (**E**,**F**) Measurement of the H_2_O_2_ and MDA contents in WT and *ls1* plants. (**G**,**H**) Activities of the ROS-related enzymes SOD and CAT in WT and *ls1* plants. The error bars indicate SDs of three biological replicates. The *p*-value was calculated using a Student’s *t*-test. ** *p* < 0.01.

**Figure 5 ijms-23-14464-f005:**
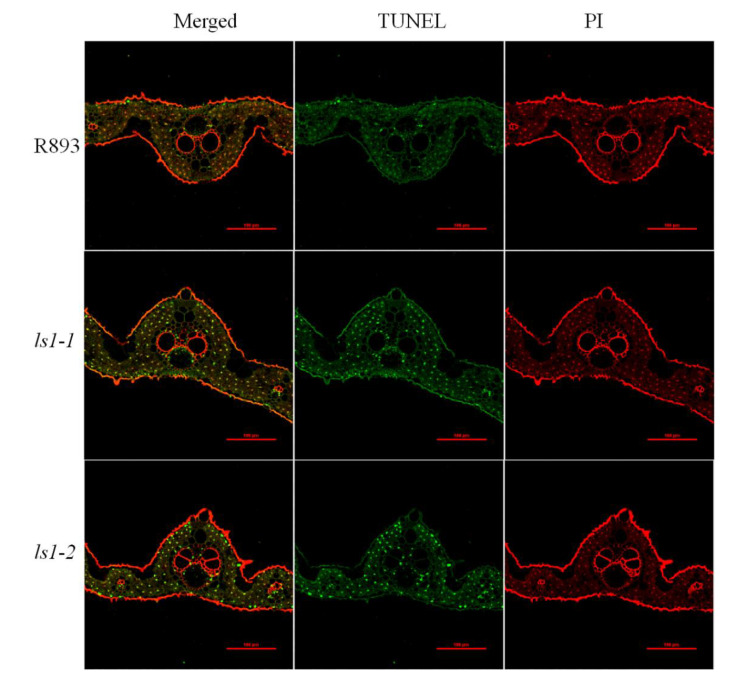
Programmed cell death (PCD) detection in the leaves of *ls1* plants. The red signal represents staining with propidium iodide (PI); yellow and green signals indicate TUNEL-positive nuclei of dead cells due to PCD. WT and *ls1* leaves were analyzed at the heading stage. The green fluorescence (520 nm) of apoptotic cells (TUNEL) in a red (620 nm) background (propidium iodide, PI) was detected with a Zeiss LSM880 confocal laser scanning microscope. Scale bar = 100 μm.

**Figure 6 ijms-23-14464-f006:**
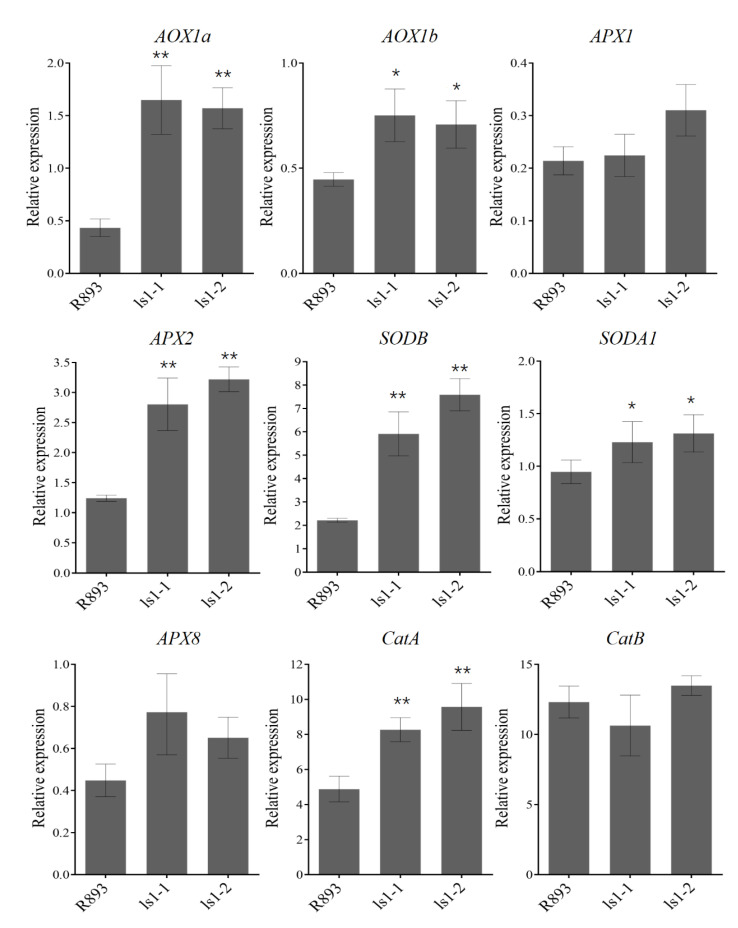
Expression levels of senescence-related genes in WT and *ls1* plants. Transcript levels of several ROS-associated marker genes were analyzed using RT-qPCR. Total RNA was obtained from the leaves of WT and *ls1* plants at the heading stage. The error bars indicate SDs of three biological replicates. The *p*-value was calculated using a Student’s *t*-test. ** *p* < 0.01, * 0.01 < *p* < 0.05.

**Figure 7 ijms-23-14464-f007:**
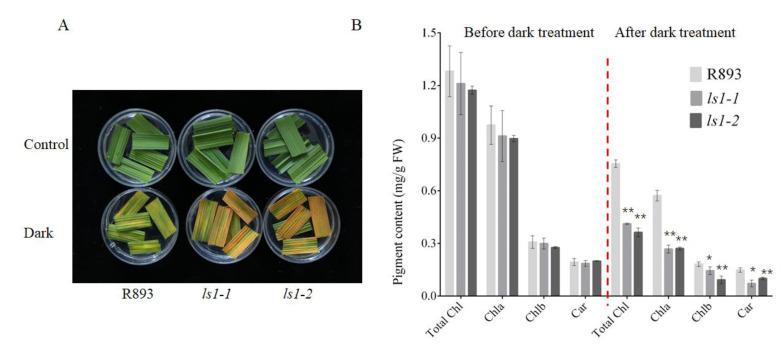
The *ls1* plants exhibited accelerated leaf yellowing during dark-induced senescence. (**A**) Detached leaves of WT and *ls1* plants were immersed in 15 mL ddH_2_O at 28 °C for 4 days in complete darkness. (**B**) Pigment contents of detached leaves of WT and *ls1* plants before and after darkness treatment. The error bars indicate SDs of three biological replicates. The *p*-value was calculated using a Student’s *t*-test. ** *p* < 0.01, * 0.01 < *p* < 0.05.

## Data Availability

The data are contained within the article or the Appendix A.

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
