# Peer review of "Mutation of Leaf Senescence 1 Encoding a C2H2 Zinc Finger Protein Induces ROS Accumulation and Accelerates Leaf Senescence in Rice"

_ijms, 2022, doi:10.3390/ijms232214464_

Round 1

Reviewer 1 Report

Authors describe a function of LS1, a rice ZF protein involved in senescence process. The experiments are well designed and well performed, with discrete points to be addressed for manuscript acceptance. Follow my suggestions over improving the quality of the manuscript aiming to attend the high standards of IJMS.

1) Results - Figure 1B

·       Do not refer the phylogenetic tree as an evolutionary tree. They are different. An evolutionary tree may encompasses the paralogous genes in rice to understand gene differentiation or, even speciation, in the rice. Hence, I suggest referring it as phylogenetic tree.

·         For discussing about the ways LS1 has followed, authors may get in evidence the bootstrap value of the analysis, as well as the statistical method used for sequencing comparison. Have authors used deduced amino acid sequences or nucleotide sequences? Put the values of branch collapses for making data interpretation more robust.

·         Do not use the code of homologous genes from database they came. Use the gene code from the reference genome (search it at Phytozome). It facilitates readers that would like to follow them.

2) Line 169 – Figure 2C call appears before Figure 2A and B – correct it – they should appear sequentially.

3) Results - Figure 4G and H and Figure 6

·         The activity of CAT is hallmarked lower in ls1 mutants, suggesting that ROS detoxification in these mutants is lower than in WT plants. However, the activity of SOD is significantly higher. In the same way, the expression of many genes encoding ROS-avoiding enzymes is much higher in the mutant. Authors may discuss these results better. How can a plant with higher ROS-avoiding related enzymes be higher and, even then, display higher ROS accumulation? There are some compensatory mechanisms whereas some of them are downregulated? Indeed, higher ROS accumulation induces the expression of enzymatic antioxidant system, but during a specific time range. To enhance the confidence of these data, authors can compare WT x ls1 plants in a timelapse (senescence onset, progression and finalization) or a kinetic experiment with some stressor that generates ROS, such as mannitol, PEG, ABA or SA.

4) Results – Figure 5

·         The pictures of TUNEL as well as PI staining are not good. Staining assays hallmark the nucleai and in the pictures, I cannot see them perfectly. Particularly, I can only see the fluorescence background.

·         Use two set of pics, one with an overview of the field (like those authors put) with better resolution, and other with higher magnification, from some representative area in which the nucleai are more evident and well stained.

·         PI also stains the cell wall of alive cells, as we can see in the pictures. Honestly, I did not see any nucleus stained in the field you’ve chosen.

·         In the caption, indicate the laser, excitation and emission lengths, besides the bypass filter you’ve used.

Reviewer 2 Report

I have few concerns about the manuscript.

1. Why authors used R893 rice varieties as WT. Its not very common to use as WT.

2. The authors reported gene expression study using qPCR. However, authors did not mentioned how they analyze the qPCR data. Which method they used SYBR Green chemistry or Taqman? Authors should provide qPCR data as supplemental file. 
3. Also, in the qPCR experiment, only one houskeeping gene used which is kind of not good practice? Is there any criteria to use specific housekeeping gene? 

4. Authors did not show the data of mRNA level of ls1 mutant. How the authors sure the LS1 gene is knocked out or level is decreased? Authors should provide either RT-PCR data or qPCR data to show the mRNA level of LS1 gene in both CRISPR mutant.

Round 2

Reviewer 2 Report

As I mentioned before. qPCR can give false results if not analyzed properly. Authors provide the qPCR data for ls1 mutant. However, I am not convinced with the data presented here. I have few concerns about the qPCR data.

1. In line no. 384 the authors mentioned each sample was repeated three times. However, in the supplemented files authors provide data is not agreement with the method section. 

2. Usually for qPCR, PCR cycle should not exceed more than 40 cycle but the supplemented data have CT value more than 40. Can you explain why?

Round 3

Reviewer 2 Report

Thank you for the addressing the concerns. Please describe the method section carefully so that readers will not get confused. Also, please provide the melt curve data for your qPCR experiment as supplemental files. Readers will have idea about the specificity of the experiments. Also its always a good idea to add, NO-RT control as well as negative control to check the specificity of the experiment. 
